# Contextual Combinatorial Multi-armed Bandits with Volatile Arms and Submodular Reward

**Lixing Chen, Jie Xu**
Department of Electrical and Computer Engineering
University of Miami
Coral Gables, FL 33146
`{lx.chen, jiexu}@miami.edu`

**Zhuo Lu**
Department of Electrical Engineering
University of South Florida
Tampa, FL 33620
`zhuolu@usf.edu`

## Abstract

In this paper, we study the stochastic contextual combinatorial multi-armed bandit (CC-MAB) framework that is tailored for volatile arms and submodular reward functions. CC-MAB inherits properties from both contextual bandit and combinatorial bandit: it aims to select a set of arms in each round based on the side information (a.k.a. context) associated with the arms. By "volatile arms", we mean that the available arms to select from in each round may change; and by "submodular rewards", we mean that the total reward achieved by selected arms is not a simple sum of individual rewards but demonstrates a feature of diminishing returns determined by the relations between selected arms (e.g. relevance and redundancy). Volatile arms and submodular rewards are often seen in many real-world applications, e.g. recommender systems and crowdsourcing, in which multi-armed bandit (MAB) based strategies are extensively applied. Although there exist works that investigate these issues separately based on standard MAB, jointly considering all these issues in a single MAB problem requires very different algorithm design and regret analysis. Our algorithm CC-MAB provides an online decision-making policy in a contextual and combinatorial bandit setting and effectively addresses the issues raised by volatile arms and submodular reward functions. The proposed algorithm is proved to achieve $O(cT^{\frac{2\alpha+D}{3\alpha+D}} \log(T))$ regret after a span of $T$ rounds. The performance of CC-MAB is evaluated by experiments conducted on a real-world crowdsourcing dataset, and the result shows that our algorithm outperforms the prior art.

## 1 Introduction

Multi-armed bandit (MAB) problems are among the most fundamental sequential decision problems with an exploration vs. exploitation trade-off. In such problems, a decision maker chooses one of several "arms", and observes a realization from an unknown reward distribution. Each decision is made based on past decisions and observed rewards. The objective is to maximize expected cumulative reward [4] over some time horizon by balancing exploration (to learn the average reward of different arms) and exploiting (to select arms that have yielded high reward in the past). The performance of a decision policy is measured by the expected regret, defined as the gap between the expected reward achieved by the algorithm and that achieved by an oracle algorithm always selecting the best arm. Contextual bandit [1] is an extension of the standard MAB framework where there is some side information (also called *context*) associated with each arm that determines the reward distribution. Contextual bandit also aims to maximize the cumulative reward by selecting one arm in each round, but now the contexts can be leveraged to predict the expected reward of arms.

However, in many real-world scenarios, e.g. recommender systems [20] and crowdsourcing [21], a decision maker needs to select multiple arms (e.g., recommended items and crowdsourcing workers) in each time slot. Such MAB problems fall into the category of *combinatorial bandit* where a set of arms rather than one individual arm are chosen in each round. What is more complicated is that the total reward of selected arms is often not a simple sum of the reward of individual arms in the set. For example, in a recommender system, recommending a diverse set of items increases the chance that a user would like at least one item. As such, recommending multiple redundant items would produce little benefit. This notion of diminishing returns due to redundancy is often captured formally using submodularity [14].

Another thorny issue in applying standard MAB frameworks in practice is the assumption on a constant set of arms that are available indefinitely. However, due to the inherent dynamic nature of many real-world applications, the arms available in each round may change dynamically over time. For example, potential crowdsourcing workers vary depending on specific tasks, location, and time. The variation of the arm set over time has been considered by MAB variants known as *sleeping bandit* [11] and *volatile bandit* [3] where arms can become available or unavailable in each round. However, these works are developed on the standard MAB framework and little effort has been made to extend volatile arms to the contextual or combinatorial MAB setting.

In this paper, we develop a novel online decision-making approach based on contextual and combinatorial bandit to address various challenges caused by volatile arms and submodular reward functions. Although our algorithm inherits concepts from some existing MAB problems, jointly considering these issues in one MAB framework requires very different algorithm design and regret analysis. The main contribution of this paper is summarized as follows: (i) We propose a contextual combinatorial multi-armed bandit algorithm (CC-MAB) framework that is compatible with submodular reward functions and volatile arms. (ii) We rigorously prove the performance guarantee of the proposed CC-MAB, which shows a $O(cT^{\frac{2\alpha+D}{3\alpha+D}}\log(T))$ regret upper bound after playing $T$ rounds. (iii) We evaluate the proposed algorithm on a real-world dataset as a crowdsourcing problem. The result shows that our approach significantly outperforms other existing MAB algorithms.

## 1.1 Difference from Existing MAB Frameworks

***Contextual bandit***: Contextual bandit considers the scenario where decision makers can observe the context of arms and infer the rewards of other unseen arms. In general, the contextual bandit framework is more applicable than the non-contextual variants, as it is rare that no context is available [15]. Various contextual bandit algorithms have been proposed, e.g. LinUCB [16], Epoch-Greedy [15] for the stochastic setting and EXP4, EXP4.P [2] for the adversarial setting. However, most of these algorithms only allow the decision makers to pull one single arm in each round. Incorporating combinatorial bandit into a context-aware setting is still an under-studied topic.

***Combinatorial bandit and submodular reward function***: Efforts have been made to generalize the classical MAB problems to combinatorial bandit [6, 8] which allows the decision makers to pull a set of arms in each round. However, most of these works are developed on standard bandit problems and neglect the available context of arms. Moreover, the submodular reward is a special issue often encountered in combinatorial bandit since the total reward of multiple selected arms may depend on the relations between individual arms. There exist works that consider submodular function in combinatorial bandit [9, 21] but they are for the non-contextual setting. Authors in [5] use a bandit framework to learn the submodular utility function. The most related work is probably [18], which investigates contextual combinatorial bandit with submodular reward. However, it considers a special submodular reward function that depends on a linear combination of individual rewards and assumes a constant set of arms which is very different from the volatile arms in our work.

***Volatile bandit and Sleeping bandit***: The key idea of volatile bandit [3] and sleeping bandit [11] is that the arms may "appear" or "disappear" in each round. Volatile and sleeping bandit are beneficial extensions of standard MAB problems and fit many practical applications where the available arms vary over time. While these works only consider volatile arms for standard MAB, our paper considers volatile arms in the contextual combinatorial MAB with submodular rewards. In addition, although volatile and sleeping bandit allow available arms to change over time, they still assume that the arms appear in each round come from an already-known finite arm set. By contrast, we allow infinitely many arms in CC-MAB by taking advantage of the Hölder (Lipschitz) condition on the context space, which also is the basis of works on *Lipschitz bandit* [12] in *continuum bandit* [13].

The rest of paper is organized as follows: Section 2 formulates a contextual combinatorial MAB problem (CC-MAB) with volatile arms and submodular functions. Section 3 introduces the algorithm design and analyzes the regret of CC-MAB. Section 4 evaluates the performance of CC-MAB on a real-world crowdsourcing problem, followed by the conclusion in Section 5.

## 2 Preliminaries and Problem Formulation

We consider a sequential decision-making process for a horizon of $T$ time slots (rounds) and formulate it as a contextual combinatorial multi-armed bandit problem. Let $\mathcal{M}^t = \{1, \dots, M^t\}$ be the set of arms arrived/available in time slot $t$. Notice that this formulation captures the volatile arms: arm sets $\mathcal{M}^t, \forall\, 0 < t \le T$ (and their size) in different time slots can be different from each other. For each arrived arm $m \in \mathcal{M}^t$, its *context* (*side information*) $x_m^t \in \mathcal{X} \triangleq [0,1]^D$ can be observed, where $D$ is the dimension of observed context vector and $\mathcal{X}$ is a bounded context space. Let $\boldsymbol{x}^t = \{x_m^t\}_{m \in \mathcal{M}^t}$ be the context set that collects the contexts of all arms in time slot $t$. The quality (i.e., the reward of choosing an arm individually) of arm $m$ is a random variable drawn from an unknown distribution parametrized by its context $x_m^t$ and we denote this random quality by $r(x_m^t)$ and its expected value by $\mu(x_m^t) = \mathbb{E}[r(x_m^t)]$. Let $\boldsymbol{r}^t = \{r(x_m^t)\}_{m \in \mathcal{M}^t}$ collect the qualities of arms arrived in time slot $t$ and $\boldsymbol{\mu}^t = \{\mu(x_m^t)\}_{m \in \mathcal{M}^t}$ collect their expected values. Given the available arms $\mathcal{M}^t$ to choose from in each time slot, our objective is to pick a subset of arms $\mathcal{S}^t \subseteq \mathcal{M}^t$ to maximize the total reward. Usually, a decision maker will have a budget $B$ that limits the maximum number of arms that can be selected, i.e., $|\mathcal{S}^t| \le B, \forall t$. We assume this budget is a constant across time slots. Nevertheless, our formulation can be easily extended to work with time-varying budgets.

As aforementioned, many reward/payoff functions we encounter in real-world applications are submodular. Therefore, we define a submodular reward function $u : 2^{\mathcal{M}^t} \to \mathbb{R}^+$ to measure the reward achieved by an arm set. Let $u(\boldsymbol{r}^t, \mathcal{S}^t)$ be the reward of selecting the arm set $\mathcal{S}^t$, its value is jointly determined by the qualities of individual arms $\{r(x_m^t)\}_{m \in \mathcal{S}^t}$ and the relations between arms that create submodularity. The considered submodular reward function is general, and is featured by the *diminishing returns property*: given the available arm set $\mathcal{M}$ and the corresponding quality $\boldsymbol{r}$ in an arbitrary time slot, for all possible arm subsets $\mathcal{S} \subseteq \mathcal{B} \subseteq \mathcal{M}$ and any arm $m \notin \mathcal{B}$, we have

$$u(\boldsymbol{r}, \{m\} \cup \mathcal{S}) - u(\boldsymbol{r}, \mathcal{S}) \ge u(\boldsymbol{r}, \{m\} \cup \mathcal{B}) - u(\boldsymbol{r}, \mathcal{B}). \tag{1}$$

We denote the marginal reward of an arm $m$ to a set $\mathcal{S}$ by $\Delta(\boldsymbol{r}, m|\mathcal{S}) \triangleq u(\boldsymbol{r}, \{m\} \cup \mathcal{S}) - u(\boldsymbol{r}, \mathcal{S})$. Moreover, we also require the reward function to be *monotone*: for all $\mathcal{S} \subseteq \mathcal{B}$, it holds that $u(\boldsymbol{r}, \mathcal{S}) \le u(\boldsymbol{r}, \mathcal{B})$. It is assumed that the reward function is revealed at the beginning of each time slot when observing the arrived arms. For example, in the spatial crowdsourcing application [19], the reward function can be determined by the overlapped sensing area of workers (arms); and in diversified information retrieval [22], the reward function can be determined by the topic of articles (arms). Our goal is selecting a subset of arms $\mathcal{S}^t \subseteq \mathcal{M}^t$ in each time slot $t$ to maximize the expected cumulative reward up to a finite time horizon $T$:

$$\max_{\mathcal{S}^1, \dots, \mathcal{S}^T} \quad \sum_{t=1}^T \mathbb{E}\left[u(\boldsymbol{r}^t, \mathcal{S}^t)\right] \tag{2a}$$

$$\text{s.t.} \ \ |\mathcal{S}^t| \le B, \ \ \mathcal{S}^t \subseteq \mathcal{M}^t, \ \ \forall t \tag{2b}$$

Obviously, the above problem can be decoupled into $T$ subproblems, one for each time slot $t$ as follow: $\max_{\mathcal{S}^t \in \mathcal{M}^t, |\mathcal{S}^t| \le B} \mathbb{E}\left[u(\boldsymbol{r}^t, \mathcal{S}^t)\right]$. Let us for now assume that for an arbitrary arm with context $x \in \mathcal{X}$, its expected quality $\mu(x) = \mathbb{E}[r(x)]$ is known a priori. Since the relations among arms are observed upon their arrival, $\mathbb{E}\left[u(\boldsymbol{r}^t, \mathcal{S}^t)\right]$ can be written as $u(\mathbb{E}\left[\boldsymbol{r}^t\right], \mathcal{S}^t) = u(\boldsymbol{\mu}^t, \mathcal{S}^t)$. Now, in each time slot $t$, we need to select a subset $\mathcal{S}^{*,t}(\boldsymbol{x}^t)$ that satisfies:

$$\mathcal{S}^{*,t}(\boldsymbol{x}^t) = \argmax_{\mathcal{S} \subseteq \mathcal{M}^t, |\mathcal{S}| \le B} u(\boldsymbol{\mu}^t, \mathcal{S}) \tag{3}$$

Clearly, if $|\mathcal{M}^t| \le B$, then we simply select all arms in $\mathcal{M}^t$. Since $\mathcal{S}^{*,t}(\boldsymbol{x}^t)$ is obtained by an omniscient oracle that knows the expected quality of all arrived arms, we call $\{\mathcal{S}^{*,t}(\boldsymbol{x}^t)\}_{t=1}^T$ the oracle solution. However, maximizing a submodular function with cardinality constraint in (3) is NP-hard. Fortunately, the greedy algorithm [17] (in Algorithm 1) offers a polynomial-time solution and guarantees to achieve no less than $(1 - 1/e)$ of the optimum as stated in the following Lemma:

---
**Algorithm 1** Greedy Algorithm
---
1: **Input**: arm set $\mathcal{M}^t$, reward vector $\boldsymbol{r}^t$, submodular reward function $u$, budget $B$.
2: **Initialization**: $\mathcal{S}_0 \leftarrow \emptyset, \tau \leftarrow 0$;
3: **while** $k \leq B$ **do**:
4:     $k = k + 1$;
5:     Depend on the expected reward $\boldsymbol{\mu}^t$, select $m_k = \arg\max_{m_k \in \mathcal{M}^t \setminus \mathcal{S}_{k-1}} \Delta(\boldsymbol{\mu}^t, \{m_k\}|\mathcal{S}_{k-1})$;
6:     $\mathcal{S}_k = \mathcal{S}_{k-1} \cup \{m_k\}$
7: **Return**: $\mathcal{S}^t = \mathcal{S}_k$
---

**Lemma 1.** *In an arbitrary time slot $t$, let $\mathcal{S}^t$ be the arm set selected by greedy algorithm and $\mathcal{S}^{*,t}$ be the optimal arm set for the problem in* (3)*, we will have $u(\boldsymbol{\mu}^t, \mathcal{S}^t) \geq (1 - \frac{1}{e})u(\boldsymbol{\mu}^t, \mathcal{S}^{*,t})$.*

The proof for Lemma 1 is omitted here, see [17] for detail. Since there is no polynomial time algorithm that achieves a better approximation in general for submodular function maximization than the greedy algorithm [7], we use it to solve the submodular function maximization problem for each per-slot subproblem in (3) and define the regret of an algorithm up to slot $T$ against this benchmark as follows:

$$R(T) = (1 - \frac{1}{e}) \cdot \sum_{t=1}^{T} \mathbb{E}\left[u(\boldsymbol{r}^t, \mathcal{S}^{*,t})\right] - \sum_{t=1}^{T} \mathbb{E}\left[u(\boldsymbol{r}^t, \mathcal{S}^t)\right] \qquad (4)$$

Here, the expectation is taken with respect to the choices made by a learning algorithm and the distributions of qualities.

## 3 Contextual Combinatorial MAB

In practice, the expected quality of an arm is unknown a priori. In this case, the per-slot subproblem cannot be solved as described previously. Therefore, we have to learn the expected qualities of arms over time using a MAB framework. However, learning the quality for volatile arms faces special challenges especially when a universal arm set $\mathcal{U}$ (i.e., $\mathcal{M}^t \subseteq \mathcal{U}$ holds true for all $t$) is not well-defined. Let us consider an extreme case that the arms arrived in every slot are completely new (i.e., the universal arm set is infinitely large), then the decision maker is unable to play an arm several times to learn the expected quality as in standard MAB algorithms and it is meaningless to do that since the arm will not appear again. To tailor our CC-MAB for a general case of volatile arms, we resort to the contextual bandit with similarity information. Basically, we divide the arms into different groups based on their context information and learn the expected quality for each group of arms by assuming that arms with similar context information will have similar qualities. This idea is also used by Lipschitz bandit [12] in continuum bandit to deal with the infinite number of arms. CC-MAB is carefully designed to properly group the volatile arms and define a control policy that makes a good trade-off between *exploration* (i.e., to learn the expected qualities of arms) and *exploitation* (i.e., to use the learned qualities to guide the future arm selection) to achieve a sublinear regret.

### 3.1 Algorithm Structure

The pseudo-code of CC-MAB is presented in Algorithm 2. Given the time horizon $T$, CC-MAB first creates a partition $\mathcal{P}_T$ which splits the context space $\mathcal{X}$ into $(h_T)^D$ hypercubes of identical size $\frac{1}{h_T} \times \cdots \times \frac{1}{h_T}$. These hypercubes correspond to possible arm groups whose expected qualities needs to be estimated. The parameter $h_T$ is a critical variable that determines the performance of CC-MAB. Its value design will be discussed in detail later. Each hypercube $p \in \mathcal{P}_T$ keeps (i) a counter $C^t(p)$ to record the number of times that an arm with $x \in p$ is chosen and (ii) a collection of observed qualities $\mathcal{E}^t(p)$ realized by selected arms with context from hypercube $p$. Then, the quality of arms with context $x \in p$ can be estimated by the sample mean $\hat{r}^t(p) = \frac{1}{C^t(p)} \sum_{r \in \mathcal{E}^t(p)} r$. Note that the collection $\mathcal{E}^t(p)$ does not appear in the algorithm, since $\hat{r}^t(p)$ can be computed based on $\hat{r}^{t-1}(p)$, $C^{t-1}(p)$, and realized qualities of selected arms in current slot $t$.

In each time slot $t$, CC-MAB performs the following steps: the context of arrived arms $\boldsymbol{x}^t = \{x_m^t\}_{m \in \mathcal{M}^t}$ is observed. For each context $x_m^t$, the algorithm determines a hypercube $p_m^t \in \mathcal{P}_T$, such

---
**Algorithm 2** CC-MAB
---
 1: **Input**: $T$, $h_T$, $K(t)$, $\mathcal{X}$.
 2: **Initialization**: context partition $\mathcal{P}_T$; set $C^0(p) = 0, \hat{r}(p) = 0, \forall p \in \mathcal{P}_T$;
 3: **for** $t = 1, \ldots, T$ **do**:
 4:     Observe arrived arms $\mathcal{M}^t$ and their contexts $\boldsymbol{x}^t = (x_m^t)_{m \in \mathcal{M}^t}$;
 5:     Find $\boldsymbol{p}^t = (p_m^t)_{m \in \mathcal{M}^t}$ such that $x_m^t \in p_m^t, p_m^t \in \mathcal{P}_T, m \in \mathcal{M}^t$;
 6:     Identify under-explored hypercubes $\mathcal{P}^{\text{ue},t}$ and the arm set $\mathcal{M}^{\text{ue},t}$; let $q = |\mathcal{M}^{\text{ue},t}|$;
 7:     **if** $\mathcal{P}^{\text{ue},t} \neq \emptyset$ **then**:                                                      ▷ *Exploration*
 8:         **if** $q \geq B$ **then** : $\mathcal{S}^t \leftarrow$ randomly pick $B$ arms from $\mathcal{M}^{\text{ue},t}$;
 9:         **else**: $\mathcal{S}^t \leftarrow$ pick $q$ arms in $\mathcal{M}^{\text{ue},t}$ and other $(B - q)$ as in (6);
10:     **else**: $\mathcal{S}^t \leftarrow$ pick $B$ arms as in (7);                                        ▷ *Exploitation*
11:     **for** each arm $m \in \mathcal{S}^t$ **do**:
12:         Observe the quality $r_m$ of arm $m$;
13:         Update reward estimation: $\hat{r}(p_m^t) = \frac{\hat{r}(p_m^t)C(p_m^t)+r_m}{C(p_m^t)+1}$;
14:         Update counters: $C(p_m^t) = C(p_m^t) + 1$;
---

that $x_m^t \in p_m^t$ holds. The collection of these hypercubes in slot $t$ is denoted by $\boldsymbol{p}^t = \{p_m^t\}_{m \in \mathcal{M}^t}$. Then the algorithm checks if there exist hypercubes $p \in \boldsymbol{p}^t$ that have not been explored sufficiently often. For this purpose, we define the *under-explored* hypercubes in slot $t$ as:

$$\mathcal{P}_T^{\text{ue},t} \triangleq \{p \in \mathcal{P}_T \mid \exists \, m \in \mathcal{M}^t, x_m^t \in p, C^t(p) \leq K(t)\} \tag{5}$$

where $K(t)$ is a deterministic, monotonically increasing control function that needs to be designed by CC-MAB. In addition, we collect the arms that fall in the under-explored hypercubes in $\mathcal{M}^{\text{ue},t} \triangleq \{m \in \mathcal{M}^t \mid p_m^t \in \mathcal{P}_T^{\text{ue},t}\}$. Depending on the under-explored arms $\mathcal{M}^{\text{ue},t}$ in time slot $t$, CC-MAB can either be in an exploration phase or exploitation phase.

If the set of under-explored arms is non-empty, i.e. $\mathcal{M}^{\text{ue},t} \neq \emptyset$, the algorithm enters an exploration phase. Let $q = |\mathcal{M}^{\text{ue},t}|$ be the size of under-explored arms. If the set of under-explored arms contains at least $B$ arms, i.e. $q \geq B$, then CC-MAB randomly selects $B$ arms from $\mathcal{M}^{\text{ue},t}$. If the under-explored arm set contains less than $B$ elements, i.e., $q < B$, then CC-MAB selects all $q$ arms from $\mathcal{M}^{\text{ue},t}$. Since the budget $B$ is not fully utilized, the rest $(B - q)$ additional arms are selected sequentially using the greedy algorithm by exploiting the estimated qualities $\hat{\boldsymbol{r}}^t$ as follows:

$$m_k = \underset{m_k \in \mathcal{M}^t \setminus \{\mathcal{M}^{\text{ue},t} \cup \mathcal{S}_{k-1}\}}{\arg\max} \Delta(\hat{\boldsymbol{r}}^t, \{m_k\} | \{\mathcal{S}_{k-1} \cup \mathcal{M}^{\text{ue},t}\}), \quad k = 1, \ldots, (B - q) \tag{6}$$

where $\mathcal{S}_{k-1} = \{m_i\}_{i=1}^{k-1}$. If the arm defined by (6) in not unique, ties are broken arbitrarily. Note that by this procedure, even in exploration phases, the algorithm exploits whenever the number of under-explored arms is smaller than the budget.

If the set of under-explored arms is empty, i.e., $\mathcal{M}^{\text{ue},t} = \emptyset$, the algorithm enters an exploitation phase. It selects all $B$ arms based on estimated qualities using the greedy algorithm:

$$m_k = \underset{m_k \in \mathcal{M}^t \setminus \mathcal{S}_{k-1}}{\arg\max} \Delta(\hat{\boldsymbol{r}}^t, \{m_k\} | \mathcal{S}_{k-1}), \quad k = 1, \ldots, B \tag{7}$$

After selecting the arm set, CC-MAB observes the qualities realized by selected arms and then updates the estimated quality and the counter of each hypercube in $\boldsymbol{p}^t$.

It remains to design the input parameter $h_T$ and the control policy $K(t)$ in order to achieve a sublinear regret in the time horizon $T$, i.e., $R(T) = O(T^\gamma)$ with $\gamma < 1$, such that CC-MAB guarantees an asymptotically optimal performance since $\lim_{T \to \infty} R(T)/T = 0$ holds.

## 3.2   Parameter Design and Regret Analysis

In this section, we design the algorithm parameters $h_T$ and $K(t)$ and give a corresponding upper bound for the regret incurred by CC-MAB. The regret analysis is carried out based on the natural assumption that the expected qualities of arms are similar if they have similar contexts. This assumption is formalized by the Hölder condition as follows:

**Assumption 1** (Hölder Condition). *There exists $L > 0$, $\alpha > 0$ such that for any two contexts $x, x' \in \mathcal{X}$, it holds that*

$$|\mu(x) - \mu(x')| \leq L\|x - x'\|^{\alpha} \tag{8}$$

*where $\|\cdot\|$ denotes the Euclidean norm in $\mathbb{R}^D$.*

Assumption 1 is needed for the regret analysis, but it should be noted that CC-MAB can also be applied if this assumption does not hold. However, a regret bound might not be guaranteed in this case. Now, we set $h_T = \lceil T^{\frac{1}{3\alpha+D}} \rceil$ (where $D$ is the dimension of the context space) for the context space partition, and $K(t) = t^{\frac{2\alpha}{3\alpha+D}} \log(t)$ in each time slot $t$ for identifying the under-explored hypercubes and arms. Then, we will have a sublinear regret upper bound of CC-MAB as follows:

**Theorem 1** (Regret Upper Bound). *Let $K(t) = t^{\frac{2\alpha}{3\alpha+D}} \log(t)$ and $h_T = \lceil T^{\frac{1}{3\alpha+D}} \rceil$. If CC-MAB is run with these parameters and Hölder condition holds true, the regret $R(T)$ is bounded by*

$$R(T) \leq (1 - \frac{1}{e}) \cdot Br^{\max}2^D \left( \log(T)T^{\frac{2\alpha+D}{3\alpha+D}} + T^{\frac{D}{3\alpha+D}} \right)$$

$$+ (1 - \frac{1}{e}) \cdot B^2 r^{\max} \binom{M^{\max}}{B} \frac{\pi^2}{3} + \left( 3BLD^{\alpha/2} + \frac{2Br^{\max} + 2BLD^{\alpha/2}}{(2\alpha+D)/(3\alpha+D)} \right) T^{\frac{2\alpha+D}{3\alpha+D}}.$$

*The leading order of the regret $R(T)$ is $O(cT^{\frac{2\alpha+D}{3\alpha+D}} \log(T))$ where $c = (1 - \frac{1}{e})Br^{\max}2^D$.*

*Proof.* See Appendix A in the supplemental file. □

The upper bound in Theorem 1 is valid for any finite time horizon, thereby providing a bound on the performance loss for any finite $T$. This can be used to characterize the convergence speed of the proposed algorithm. The leading order of regret upper bound $R(T)$ mainly depends on the context dimension $D$. The role of $D$ here is similar to the role of *the number of arms* in standard MAB algorithms, e.g. UCB1[4]. Since the arm set considered in CC-MAB is infinitely large, CC-MAB splits these arms into different groups (i.e., *hypercubes*) based on their context information. The constant $(1 - \frac{1}{e})Br^{\max}2^D$ grows exponentially with the context dimension $D$, and hence regret tends to be high when the context space is large. However, a learner may apply dimension reduction techniques, e.g., feature selection, based on empirical experience to cut down the context space. In addition, the form of upper regret bound for CC-MAB is very different from that for many existing MAB algorithms, e.g. contextual/combinatorial/volatile MAB, which are developed on a finite arm set. The continuum-armed bandits (CAB), considering a continuum arm set $X \in [0, 1]$, provides a regret upper bound $O(cT^{\frac{2}{3}} \log^{\frac{1}{3}}(T))$ with fixed discretization [13]. If CC-MAB is run with one-dimension context space and the parameter $\alpha$ large enough, its regret upper bound reduces to $cT^{\frac{2}{3}} \log(T)$, which is slightly looser than the CAB.

We note that the regret bound, which although is sublinear in $T$, is loose when the budget $B$ is close to $M^{\max}$ (the maximum number of arms arriving in each time slot). Consider the special case of $B = M^{\max}$, CC-MAB is actually identical to the oracle algorithm, i.e., choosing all arms arriving in each time slot and hence the regret is 0. It is intuitive that when the budget $B$ is large, learning is not very much needed and hence the more challenging regime is when the budget $B$ is small. Furthermore, if we are able to know the specific stochastic pattern of the arm arrival, a sharper regret bound can be derived. For example, consider a case that the number of available arms in each time slot $\{M^t\}_{t=1}^{T}$ ($M^t = |\mathcal{M}^t|$) are i.i.d. random variables and the probability $\mathbb{E}[\Pr(B < M^t)] = \beta$, where $\beta \in [0, 1]$, is revealed. Then, the regret upper bound of CC-MAB can be derived in the following corollary.

**Corollary 1.** *If Assumption 1 and $\mathbb{E}[\Pr(B < M^t)] = \beta$ hold true, and CC-MAB is run with the parameters defined in Theorem 1. Let $R^{ub}$ be the regret upper bound defined in Theorem 1, the regret $R(T)$ is bounded by $R(T) \leq \beta R^{ub}$.*

Compared to Theorem 1, the regret is scaled by the parameter $\beta$ in this case. This is due to the fact that with probability $(1 - \beta)$ the budget is larger than the number of arrived arms and hence, both CC-MAB and oracle algorithm select all the arrived arms and no regret is incurred.

### 3.3 Complexity Analysis

The computational complexity of CC-MAB is mainly determined by the counters $C(p)$ and estimated context-specific qualities $\hat{r}(p)$ for each hypercube $p \in \mathcal{P}_T$ kept by the learner. If CC-MAB is run with the parameters in Theorem 1, the number of hypercubes is $(h_T)^D = \lceil T^{\frac{1}{3\alpha+D}} \rceil^D$. Hence, the required memory is sublinear in the time horizon $T$. However, this also means that when $T \to \infty$, the algorithm would require infinite memory. Fortunately, in the practical implementations, A learner only needs to keep the counters of hypercubes $p$ to which at least one of appeared arms' context vectors belongs. Hence the required number of counters that have to be kept is actually much smaller than the analytical requirement. Moreover, the learner may choose to stop splitting the context space after a certain level of granularity so the number of hypercubes will be bounded.

## 4 Experiments

### 4.1 Experiment setting

We evaluate the performance of CC-MAB in a crowdsourcing application based on the data published by Yelp[1]. The dataset provides abundant real-world traces for emulating spatial crowdsourcing tasks where Yelp users are assigned with tasks to review local businesses. The dataset contains 61,184 businesses, 36,6715 users and 1,569,264 reviews.

*1) Arms and Context*: We divide the time span of the dataset into daily instances, and every day (each time slot) a set of users are selected to review businesses. In each time slot, a user can choose one business from a set of reachable businesses to review. Therefore, each user-business pair is an arm in our crowdsourcing problem. Note that the available users vary across the time and the businesses a user can review also change depending on users' location. This means that the available arms in each round are different. In addition, each user has side information including the number of fans, the number of received votes, and the years a user was elite, which can be used as arm context.

*2) System Reward*: When soliciting reviews on businesses, the decision maker expects more businesses covered under the budget constraint. Therefore, individual user's marginal quality is maximized if each of them reviews a distinct business. In the case that a business is reviewed more than once, the value of subsequent reviews will be discounted. In order to capture this property, we use Dixit-Stiglitz preference model [23] to calculate the total reward. Let $r_{ij}$ denote the $j$-th review on the $i$-th business. The quality of each review record (arm) is calculated based on the text length of the comment, and the number of votes it received ($r_{ij} = L_{ij} N_{ij}^{\text{votes}}$ where $L_{ij}$ is the normalized text length and $N_{ij}^{\text{votes}}$ is the vote count). Figure 1 shows the distribution of arm quality over the *number of fans* context. We see that the reviews from users with a larger number of fans tend to have a higher quality. Figure 2 further depicts the expected quality of user groups in each hypercube on two context dimensions *number of years as elite* and *number of fans*, which are used to define the context space in the experiment. We see that the review quality is very related to the users' context.

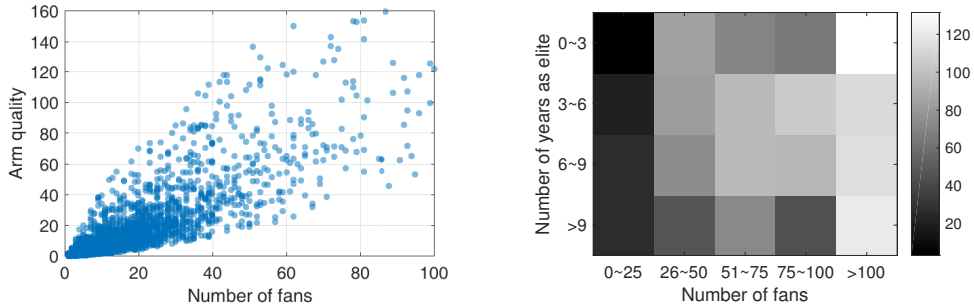

Figure 1: Arm quality distribution on *fans* context.    Figure 2: Expected quality for hypercubes.

For each business $i$, its reward is given by the Dixit-Stiglitz model:

$$u_i = \left(\sum\nolimits_{j} (r_{ij})^p\right)^{1/p}, \quad p \geq 1. \tag{9}$$

[1]Yelp dataset challenge: www.yelp.com/dataset/challenge

and the total reward is the sum of rewards of all businesses: $u = \sum_i u_i$. It can be easily verified that the given reward function is submodular for any value of parameter $p \geq 1$. We compare CC-MAB with the following benchmarks:

**1) Oracle**: Oracle knows precisely the expected quality of each arm. In each time slot, Oracle chooses $B$ arms using the greedy algorithm presented in Algorithm 1.

**2) $k$-LinUCB**: LinUCB [16] is a contextual bandit algorithm which recommends exactly one arm in each round. To select a set of $k$ users, we repeat LinUCB algorithm $k$ times in each round. By sequentially removing selected arms, we ensure that the $k$ arms returned by $k$-LinUCB are distinct in each round. Notice that $k$-LinUCB is unaware of the submodular reward when selecting users.

**3) UCB**: UCB algorithm [4] is a classical MAB algorithm (non-contextual and non-combinatorial) that achieves the logarithmic regret bound. Similar to $k$-LinUCB, we repeat UCB $k$ times to select multiple users in each round.

**4) CC-MAB-NS**: CC-MAB-NS(Non-Submodular) is a variant of proposed CC-MAB where submodularity of reward function is not considered. It simply selects $B$ (or $(B - k)$) arms with the highest quality during exploitation (or semi-exploitation).

**5) Random**: The Random algorithm picks $B$ arms randomly from the available arms in each round.

### 4.2   Results and Discussions

Figure 3 shows the cumulative system rewards achieved by CC-MAB and 5 benchmark algorithms. As expected, Oracle has the highest cumulative reward and gives an upper bound to other algorithms. Among other algorithms, we see that the context-aware algorithms CC-MAB, $k$-LinUCB and CC-MAB-NS outperform UCB and Random algorithm. This indicates that exploiting the context information of arms helps to better learn the quality of arms. Further, it can be observed that the CC-MAB achieves a close-to-oracle performance while $k$-LinUCB and CC-MAB-NS incur obvious reward loss since they do not consider the submodularity of the reward function.

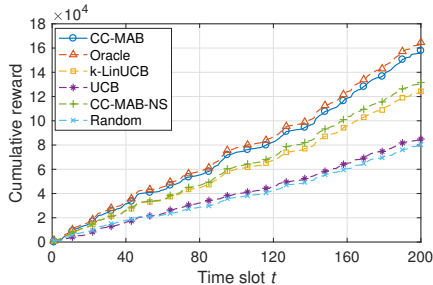

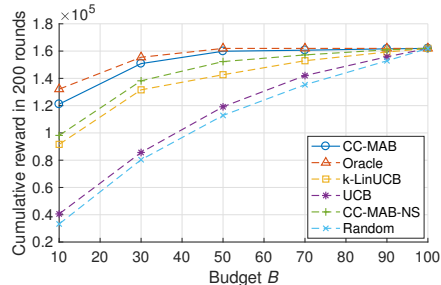

Figure 3: Comparison of cumulative rewards.    Figure 4: Cumulative rewards over budgets.

Figure 4 shows the cumulative rewards achieved by 6 algorithms in 200 rounds under different budgets. In general, all the algorithms achieve a higher cumulative reward with a larger budget since more arms can be selected. It is worth noticing that the cumulative rewards obtained by Oracle and CC-MAB become saturated at budget $B = 50$ which is much smaller compared to that of other algorithms that become saturated after $B = 90$. This is due to the fact that the most beneficial arms can be efficiently identified by Oracle and CC-MAB algorithms by considering the submodularity of the reward function, and therefore the arms that are left out only offer little marginal reward. Note that, in our experiment, the maximum number of arms in each round is set as 100 and hence all algorithms have the same performance at $B = 100$. In addition, we see that CC-MAB is able to achieve the close-to-oracle performance at all budget levels. In Figure 5, we further depict the regret of CC-MAB across different budgets. It shows that the regret decreases with the increase in budget. This seems contradictory to Theorem 1 where the regret upper bound grows with the budget $B$. However, the regret upper bound is proved for an arbitrary submodular function and an arbitrarily large arm set in each round. In a setting that the maximum number of arms arriving in each round is fixed, increasing the budget reduces the chance that beneficial arms are left out.

We also analyze the impact of submodularity to our algorithm. Figure 6 shows the cumulative reward achieved by CC-MAB and CC-MAB-NS with different levels of submodularity, where the level of submodularity is determined by the parameter $p$ in the reward function (9). A larger $p$ indicates a stronger submodularity. When $p = 1$, the reward function becomes non-submodular, therefore CC-

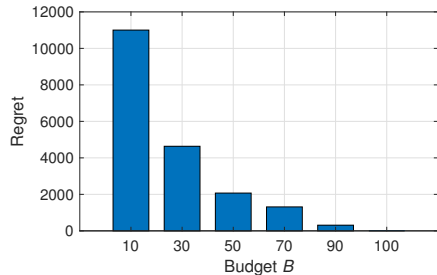

Figure 5: Regret of CC-MAB over budgets.

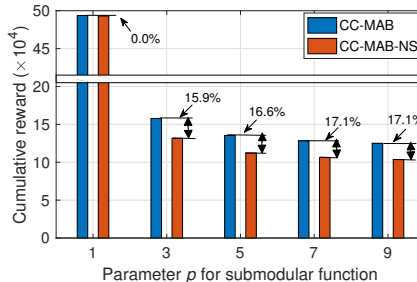

Figure 6: Impact of submodularity.

MAB and CC-MAB-NS achieve the same cumulative reward. Moreover, at this point, the achieved reward is maximized since the *diminishing return* of submodularity disappears. When the reward function becomes submodular ($p > 1$), CC-MAB outperforms CC-MAB-NS, and the performance loss incurred by CC-MAB-NS grows as the submodularity becomes stronger.

Other extended simulations are given in Appendix B in the supplemental file.

## 5  Conclusion

We presented a framework called contextual combinatorial multi-armed bandit that accommodates combinatorial nature of contextual arms. An efficient algorithm CC-MAB was proposed, which is tailored to volatile arms and submodular reward functions. We rigorously proved that the regret upper bound of the proposed algorithm is sublinear in the time horizon $T$. Experiments on real-world crowdsourcing data demonstrated that our algorithm helps to explore and exploit arms' reward by considering the context information and the submodularity of reward function and hence improves the cumulative reward compared to many existing MAB algorithms. CC-MAB currently creates static context partitions during initialization which may be inappropriate in certain cases, a meaningful extension is to generate appropriate partitions dynamically over time based on the distribution of arrived arms on the context space.

## 6  Acknowledgment

L. Chen and J. Xu's work is supported in part by the Army Research Office under Grant W911NF-18-1-0343. The views and conclusions contained in this document are those of the authors and should not be interpreted as representing the official policies, either expressed or implied, of the Army Research Office or the U.S. Government. The U.S. Government is authorized to reproduce and distribute reprints for Government purposes notwithstanding any copyright notation herein.

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
