[Supplementary Material]

# A  Proof of Theorem 1

Before proceeding, we first define some auxiliary variables. For each hypercube $p \in \mathcal{P}_T$, we define $\bar{\mu}(p) = \sup_{x \in p} \mu(x)$ and $\underline{\mu}(p) = \inf_{x \in p} \mu(x)$ be the best and worst expected quality over all contexts $x \in p$. In some steps of the proofs, we need to compare the qualities at different positions in a hypercube. As a point of reference, we define the context at (geometrical) center of a hypercube $p$ as $\tilde{x}_p$ and its expected quality $\tilde{\mu}(p) = \mu(\tilde{x}_p)$. Given $\mathcal{M}^t$, $\boldsymbol{x}^t$, $\boldsymbol{p}^t$ in each time slot, let $\bar{\boldsymbol{\mu}}_p^t = [\bar{\mu}(p_1^t), \ldots, \bar{\mu}(p_{M^t}^t)]$, $\underline{\boldsymbol{\mu}}_p^t = [\underline{\mu}(p_1^t), \ldots, \underline{\mu}(p_{M^t}^t)]$, $\tilde{\boldsymbol{\mu}}_p^t = [\tilde{\mu}(p_1^t), \ldots, \tilde{\mu}(p_{M^t}^t)]$, and define $\tilde{\mathcal{S}}^{*,t}(\boldsymbol{p}^t)$

$$\tilde{\mathcal{S}}^{*,t}(\boldsymbol{p}^t) = \underset{\mathcal{S} \subseteq \mathcal{M}^t, |\mathcal{S}| \leq b}{\arg\max} \; u(\tilde{\boldsymbol{\mu}}_p^t, \mathcal{S}) \tag{10}$$

Let $\tilde{\mathcal{S}}^t(\boldsymbol{p}^t)$ be the greedy optimal set in polynomial time to the problem in (10) and we will have $u(\tilde{\boldsymbol{\mu}}_p^t, \tilde{\mathcal{S}}^t(\boldsymbol{p}^t)) \geq (1 - 1/e) \cdot u(\tilde{\boldsymbol{\mu}}_p^t, \tilde{\mathcal{S}}^{*,t}(\boldsymbol{p}^t))$. The arm set $\tilde{\mathcal{S}}^t(\boldsymbol{p}^t)$ is used to identify subsets of arms which are bad to select. Let

$$\mathcal{L}^t(\boldsymbol{p}^t) = \left\{ G \subseteq \mathcal{M}^t, |G| = B : u(\underline{\boldsymbol{\mu}}_p^t, \tilde{\mathcal{S}}^t(\boldsymbol{p}^t)) - u(\bar{\boldsymbol{\mu}}_p^t, G) \geq At^\theta \right\} \tag{11}$$

be the *set of suboptimal subsets of arms* for hypercubes $\boldsymbol{p}^t$, where $A > 0$ and $\theta < 0$ are parameters used only in the regret analysis. We call a subset $G$ of arms in $\mathcal{L}^t(\boldsymbol{p}^t)$ *suboptimal* for $\boldsymbol{p}^t$, since the sum of the worst expected reward in $\tilde{\mathcal{S}}^t(\boldsymbol{p}^t)$ is at least an amount $At^\theta$ higher than the sum of the best expected reward for subset $G$. We call subsets in $\mathcal{S}_B^t \backslash \mathcal{L}^t(\boldsymbol{p}^t)$ *near-optimal* for $\boldsymbol{p}^t$. Here, $\mathcal{S}_B^t$ denotes the set of all $B$-element subsets of arm set $\mathcal{M}^t$. Then, the regret $R(T)$ can be divided into the following three summands:

$$R(T) = \mathbb{E}[R_e(T)] + \mathbb{E}[R_s(T)] + \mathbb{E}[R_n(T)] \tag{12}$$

where the term $\mathbb{E}[R_e(T)]$ is the regret due to exploration phases and the terms $\mathbb{E}[R_s(T)]$ and $\mathbb{E}[R_n(T)]$ both correspond to regret in exploitation phases: $\mathbb{E}[R_s(T)]$ is the regret due to suboptimal choices, i.e., the subsets of arms from $\mathcal{L}^t(\boldsymbol{p}^t)$ are selected; $\mathbb{E}[R_n(T)]$ is the regret due to near-optimal choices, i.e., the subsets of arms from $\mathcal{S}_B \backslash \mathcal{L}^t(\boldsymbol{p}^t)$ are selected. In the following, we prove that each of the three summands is bounded.

We first give a bound of $\mathbb{E}[R_e(T)]$, which depends on the choice of two parameters $z$ and $\gamma$.

**Lemma 2** (Bound for $\mathbb{E}(R_e(T))$). *Let* $K(t) = t^z \log(t)$ *and* $h_T = \lceil T^\gamma \rceil$, *where* $0 < z < 1$ *and* $0 < \gamma < \frac{1}{D}$. *If the algorithm is run with these parameters, the regret* $\mathbb{E}[R_e(T)]$ *is bounded by*

$$\mathbb{E}[R_e(T)] \leq (1 - \frac{1}{e})Br^{\max}2^D \left( T^{z + \gamma D} \log(T) + T^{\gamma D} \right) \tag{13}$$

*Proof of Lemma 2.* Let $t$ be an exploration phase and let $\boldsymbol{p}^t = (p_m^t)_{m \in \mathcal{M}^t}$ be the hypercubes of arrived arms. Then, according to the proposed algorithm, the set of under-explored hypercubes $\mathcal{P}_T^{\mathrm{ue},t}$ is non-empty, i.e., there exists at least one arm with context $x_m^t$, such that a hypercube $p$ satisfying $x_m^t \in p$ has $C^t(p) \leq K(t) = t^z \log(t)$. Clearly, there can be at most $\lceil T^z \log(T) \rceil$ exploration phases in which arms in $p$ are selected due to under-exploration of $p$. Since there are $(h_T)^D$ hypercubes in the partition, there can be at most $(h_T)^D \lceil T^z \log(T) \rceil$ exploration phases. Notice the random quality $r(x_m^t)$ for any $m \in \mathcal{M}^t, x_m^t \in [0,1]^D$, is bounded by $r^{\max}$ and the maximum number of selected arms in each time slot is $B$. Considering the submodularity of reward function, the maximum regret of wrong selection in one exploration phase is bounded by $(1 - 1/e)Br^{\max}$. Additionally, we have to take into account the loss due to exploitations in the case that the size of $\mathcal{M}^{\mathrm{ue},t}$ is smaller than $B$. In each of these exploration phases, the maximum additional regret is $(1 - 1/e)(B - 1)r^{\max}$. Therefore, we have

$$\mathbb{E}[R_e(T)] \leq (1 - 1/e)Br^{\max}(h_T)^D \lceil T^z \log(T) \rceil$$
$$= (1 - 1/e)Br^{\max}\lceil T^\gamma \rceil^D \lceil T^z \log(T) \rceil$$

Using $\lceil T^\gamma \rceil^D \leq (2T^\gamma)^D = 2^D T^{\gamma D}$, it holds

$$\mathbb{E}[R_e(T)] \leq (1 - 1/e)Br^{\max}2^D T^{\gamma D} \left( T^z \log(T) + 1 \right)$$
$$= (1 - 1/e)Br^{\max}2^D \left( T^{z + \gamma D} \log(T) + T^{\gamma D} \right) \tag{14}$$

$\square$

Next, we give a bound for $\mathbb{E}[R_s(T)]$. This bound also depends on the choice of two parameters $z$ and $\gamma$. Additionally, a condition on these parameters has to be satisfied.

**Lemma 3** (Bound for $\mathbb{E}(R_s(T))$)**.** *Let* $K(t) = t^z \log(t)$ *and* $h_T = \lceil T^\gamma \rceil$, *where* $0 < z < 1$ *and* $0 < \gamma < \frac{1}{D}$. *If the algorithm is run with these parameters, Assumption 1 holds true and the additional condition* $2H(t) + 2BLD^{\frac{\alpha}{2}} h_T^{-\alpha} \leq At^\theta$ *is satisfied for all* $1 \leq t \leq T$ *where* $H(t) = Br^{\max} t^{-\frac{z}{2}}$, *the regret* $\mathbb{E}[R_s(T)]$ *is bounded by*

$$\mathbb{E}[R_s(T)] \leq (1 - \frac{1}{e})B^2 r^{\max} \binom{M^{\max}}{B} \frac{\pi^2}{3} \tag{15}$$

*Proof of Lemma 3.* For $1 \leq t \leq T$, let $W^t = \{\mathcal{P}^{\text{ue},t} = \emptyset\}$ be the even that slot $t$ is an exploitation phase. By the definition of $\mathcal{P}^{\text{ue},t}$, in this case, it holds that $C^t(p_m^t) > K(t) = t^z \log(t), \forall p \in \boldsymbol{p}^t$. Let $V_G^t$ be the event that subset $G \in \mathcal{L}^t(\boldsymbol{p}^t)$ is selected at time slot $t$. Then, it holds that

$$R_s(T) = \sum_{t=1}^{T} \sum_{G \in \mathcal{L}^t(\boldsymbol{p}^t)} I_{\{V_G^t, W^t\}} \times \left( (1 - \frac{1}{e})u\left(\boldsymbol{r}^t, \mathcal{S}^{*,t}(\boldsymbol{x}^t)\right) - u\left(\boldsymbol{r}^t, G\right) \right) \tag{16}$$

where, in each time slot, the loss due to selecting a suboptimal subset $G \in \mathcal{L}^t(\boldsymbol{p}^t)$ is considered. Since the maximum regret of selecting $G$ is bounded by $(1 - \frac{1}{e})Br^{\max}$, we have

$$R_s(T) \leq (1 - \frac{1}{e})Br^{\max} \sum_{t=1}^{T} \sum_{G \in \mathcal{L}^t(\boldsymbol{p}^t)} I_{\{V_G^t, W^t\}}, \tag{17}$$

and taking the exception, the regret is hence bounded by

$$\mathbb{E}[R_s(T)] \leq (1 - \frac{1}{e})Br^{\max} \sum_{t=1}^{T} \sum_{G \in \mathcal{L}^t(\boldsymbol{p}^t)} \mathbb{E}\left[I_{\{V_G^t, W^t\}}\right]$$

$$= (1 - \frac{1}{e})Br^{\max} \sum_{t=1}^{T} \sum_{G \in \mathcal{L}^t(\boldsymbol{p}^t)} \text{Prob}\left\{V_G^t, W^t\right\} \tag{18}$$

In the event of $V_G^t$, by the design of the algorithm, this means that with the estimated arm quality, the rewards of selecting arms in $G$ is at least as high as the reward of selecting arms in $\tilde{\mathcal{S}}^t(\boldsymbol{p}^t)$, i.e., $u(\hat{\boldsymbol{r}}_p^t, G) \geq u(\hat{\boldsymbol{r}}_p^t, \tilde{\mathcal{S}}^t(\boldsymbol{p}^t))$. Thus, we have:

$$\text{Prob}\left\{V_G^t, W^t\right\} \leq \text{Prob}\left\{u(\hat{\boldsymbol{r}}_p^t, G) \geq u(\hat{\boldsymbol{r}}_p^t, \tilde{\mathcal{S}}^t(\boldsymbol{p}^t))\right\} \tag{19}$$

The event in the right-hand side of (19) implies at lease one of the three following events for any $H(t) > 0$:

$$E_1 = \left\{u(\hat{\boldsymbol{r}}_p^t, G) \geq u(\bar{\boldsymbol{\mu}}_p^t, G) + H(t), W^t\right\}$$

$$E_2 = \left\{u(\hat{\boldsymbol{r}}_p^t, \tilde{\mathcal{S}}^t(\boldsymbol{p}^t)) \leq u(\boldsymbol{\mu}_p^t, \tilde{\mathcal{S}}^t(\boldsymbol{p}^t)) - H(t), W^t\right\}$$

$$E_3 = \left\{u(\hat{\boldsymbol{r}}_p^t, G) \geq u(\hat{\boldsymbol{r}}_p^t, \tilde{\mathcal{S}}^t(\boldsymbol{p}^t)), u(\hat{\boldsymbol{r}}_p^t, G) < u(\bar{\boldsymbol{\mu}}_p^t, G) + H(t), \right.$$
$$\left. u(\hat{\boldsymbol{r}}_p^t, \tilde{\mathcal{S}}^t(\boldsymbol{p}^t)) > u(\boldsymbol{\mu}_p^t, \tilde{\mathcal{S}}^t(\boldsymbol{p}^t)) - H(t), W^t\right\}.$$

Hence, we have for the original event in (19)

$$\left\{u(\hat{\boldsymbol{r}}_p^t, G) \geq u(\hat{\boldsymbol{r}}_p^t, \tilde{\mathcal{S}}^t(\boldsymbol{p}^t))\right\} \subseteq E_1 \cup E_2 \cup E_3 \tag{20}$$

The probability of the three event $E_1$, $E_2$, and $E_3$ will be bounded separately. Let start by bounding $E_1$. Recall that the best expected quality of arms in set $p$ is $\bar{\mu}(p) = \sup_{x \in p} \bar{\mu}(x)$. Therefore, the expected quality of arm $m$ in $G$ is bounded by

$$
\begin{aligned}
\mathbb{E}\left[\hat{r}(p_m^t)\right] =& \mathbb{E}\left[\frac{1}{|\mathcal{E}^t(p_m^t)|} \sum_{(\tau,k):x_k^\tau \in p_m^\tau, n \in \mathcal{S}^\tau} r(x_k^\tau)\right] \\
=& \frac{1}{|\mathcal{E}^t(p_m^t)|} \underbrace{\sum_{(\tau,k):x_k^\tau \in p_m^\tau, n \in \mathcal{S}^\tau} \underbrace{\mu(x_k^\tau)}_{\leq \bar{\mu}(p_m^t)}}_{|\mathcal{E}^t(p_m^t)|\text{summands}} \\
\leq& \bar{\mu}(p_m^t)
\end{aligned}
\tag{21}
$$

This implies

$$
\begin{aligned}
\text{Prob}\{E_1\} =& \text{Prob}\left\{u(\hat{\boldsymbol{r}}_p^t, G) \geq u(\bar{\boldsymbol{\mu}}_p^t, G) + H(t), W^t\right\} \\
\leq& \text{Prob}\left\{\hat{r}(p_m^t) \geq \bar{\mu}(p_m^t) + \frac{H(t)}{B}, \exists m \in G, W^t\right\} \\
\leq& \text{Prob}\left\{\hat{r}(p_m^t) \geq \mathbb{E}\left[\hat{r}(p_m^t)\right] + \frac{H(t)}{B}, \exists m \in G, W^t\right\} \\
=& \sum_{m \in G} \text{Prob}\left\{\hat{r}(p_m^t) \geq \mathbb{E}\left[\hat{r}(p_m^t)\right] + \frac{H(t)}{B}, W^t\right\}
\end{aligned}
$$

The first inequality comes from the fact that $\left\{u(\hat{\boldsymbol{r}}_p^t, G) \geq u(\bar{\boldsymbol{\mu}}_p^t, G) + H(t)\right\} \subseteq \left\{\hat{r}(p_m^t) \geq \bar{\mu}(p_m^t) + \frac{H(t)}{B}, \exists m \in G\right\}$, which can be easily verified by *reductio ad absurdum* and submodularity property. Now, applying Chernoff-Hoeffding bound [10] (note that for each arm, the estimated quality is bounded by $r^{\max}$) and exploiting that event $W^t$ implies that at least $t^z \log(t)$ samples were drawn, we get

$$
\begin{aligned}
\text{Prob}\{E_1\} \leq& \sum_{m \in G} \text{Prob}\left\{\hat{r}(p_m^t) - \mathbb{E}\left[\hat{r}(p_m^t)\right] \geq \frac{H(t)}{B}, W(t)\right\} \\
\leq& \sum_{m \in G} \exp\left(\frac{-2|\mathcal{E}^t(p_m^t)|H(t)^2}{B^2(r^{\max})^2}\right) \\
\leq& \sum_{m \in G} \exp\left(\frac{-2H(t)^2 t^z \log(t)}{B^2(r^{\max})^2}\right)
\end{aligned}
\tag{22}
$$

Analogously, it can be proven for event $E_2$, that

$$
\begin{aligned}
\text{Prob}\{E_2\} =& \text{Prob}\left\{u(\hat{\boldsymbol{r}}_p^t, \tilde{\mathcal{S}}^t(\boldsymbol{p}^t)) \geq u(\underline{\boldsymbol{\mu}}_p^t, \tilde{\mathcal{S}}^t(\boldsymbol{p}^t)) - H(t), W^t\right\} \\
\leq& \sum_{m \in \tilde{\mathcal{S}}^t(\boldsymbol{p}^t)} \exp\left(\frac{-2H(t)^2 t^z \log(t)}{B^2(r^{\max})^2}\right)
\end{aligned}
\tag{23}
$$

To bound the event $E_3$, we first make some additional definitions. First, we rewrite the estimate $\hat{r}(p), p \in \mathcal{P}_T$ as follows:

$$
\begin{aligned}
\hat{r}(p) =& \frac{1}{|\mathcal{E}^t(p)|} \sum_{(\tau,k):x_k^\tau \in p, m \in \mathcal{S}^t} r(x_k^\tau) \\
=& \frac{1}{|\mathcal{E}^t(p)|} \sum_{(\tau,k):x_k^\tau \in p, m \in \mathcal{S}^t} \mu(x_k^\tau) + \epsilon_k^\tau
\end{aligned}
$$

where $\epsilon_k^\tau$ denotes the deviation from the expected quality of an arm $k \in \mathcal{M}^\tau$ with context $x_k^\tau$. Additionally, we define the best and worst context for a hypercube $p \in \mathcal{P}_T$ as $x^{\text{best}}(p) \triangleq \arg\max_{x \in p} \mu(x)$

and $x^{\text{worst}}(p) \triangleq \arg\min_{x \in p} \mu(x)$, respectively. Finally, we define the best and worst quality of an arm in hypercube $p$ as

$$r^{\text{best}}(p) = \frac{1}{|\mathcal{E}^t(p)|} \sum_{(\tau,k):x_k^\tau \in p, m \in \mathcal{S}^t} \mu(x^{\text{best}}(p)) + \epsilon_k^\tau \tag{24}$$

$$r^{\text{worst}}(p) = \frac{1}{|\mathcal{E}^t(p)|} \sum_{(\tau,k):x_k^\tau \in p, m \in \mathcal{S}^t} \mu(x^{\text{worst}}(p)) + \epsilon_k^\tau \tag{25}$$

By Hölder condition from Assumption 1, since $x^{\text{best}}(p) \in p$ and only contexts from hypercube $p$ are used for calculating the estimated quality $\hat{r}(p)$, it can be shown that

$$r^{\text{best}}(p) - \hat{r}(p) \le LD^{\frac{\alpha}{2}} h_T^{-\alpha} \tag{26}$$

holds. Analogously, we have

$$\hat{r}(p) - r^{\text{worst}}(p) \le LD^{\frac{\alpha}{2}} h_T^{-\alpha} \tag{27}$$

We apply (26) and (27) to arms in $G$ and $\tilde{\mathcal{S}}^t(\boldsymbol{p}^t)$. Consider the arm set sequentially selected by greedy algorithm, for any $k \le B$, we have the marginal gain $\Delta(\boldsymbol{r}_p^{\text{best},t}, \{m_k\}|\mathcal{S}_{k-1}) \le \Delta(\hat{\boldsymbol{r}}_p^t, \{m_k\}|\mathcal{S}_{k-1}) + LD^{\frac{\alpha}{2}} h_T^{-\alpha}$. Therefore

$$u(\boldsymbol{r}_p^{\text{best},t}, G) - u(\hat{\boldsymbol{r}}_p^t, G) \le \sum_{m \in G} \left( r^{\text{best}}(p_m^t) - \hat{r}(p_m^t) \right) \le BLD^{\frac{\alpha}{2}} h_T^{-\alpha} \tag{28}$$

and summing over the $\tilde{\mathcal{S}}^t(\boldsymbol{p}^t)$ yields

$$u(\hat{\boldsymbol{r}}_p^t, \tilde{\mathcal{S}}^t(\boldsymbol{p}^t)) - u(\boldsymbol{r}_p^{\text{worst},t}, \tilde{\mathcal{S}}^t(\boldsymbol{p}^t)) \le \sum_{m \in \tilde{\mathcal{S}}^t(\boldsymbol{p}^t)} \left( \hat{r}(p_m^t) - r^{\text{worst}}(p_m^t) \right) \le BLD^{\frac{\alpha}{2}} h_T^{-\alpha} \tag{29}$$

Now the three components of event $E_3$ are considered separately. By definition of $r^{\text{best}}(p)$ and $r^{\text{worst}}(p)$ in (24) and (25). The first component of $E_3$ holds that

$$\left\{ u(\hat{\boldsymbol{r}}_p^t, G) \ge u(\hat{\boldsymbol{r}}_p^t, \tilde{\mathcal{S}}^t(\boldsymbol{p}^t)) \right\} \subseteq \left\{ u(\boldsymbol{r}_p^{\text{best},t}, G) \ge u(\boldsymbol{r}_p^{\text{worst},t}, \tilde{\mathcal{S}}^t(\boldsymbol{p}^t)) \right\} \tag{30}$$

For the second component, using (28), we have

$$\left\{ u(\hat{\boldsymbol{r}}_p^t, G) < u(\bar{\boldsymbol{\mu}}_p^t, G) + H(t) \right\}$$
$$\subseteq \left\{ u(\boldsymbol{r}_p^{\text{best},t}, G) - BLD^{\frac{\alpha}{2}} h_T^{-\alpha} < u(\bar{\boldsymbol{\mu}}_p^t, G) + H(t) \right\}$$
$$= \left\{ u(\boldsymbol{r}_p^{\text{best},t}, G) < u(\bar{\boldsymbol{\mu}}_p^t, G) + BLD^{\frac{\alpha}{2}} h_T^{-\alpha} + H(t) \right\} \tag{31}$$

For the third component, we have

$$\left\{ u(\hat{\boldsymbol{r}}_p^t, \tilde{\mathcal{S}}^t(\boldsymbol{p}^t)) > u(\underline{\boldsymbol{\mu}}_p^t, \tilde{\mathcal{S}}^t(\boldsymbol{p}^t)) - H(t) \right\}$$
$$\subseteq \left\{ u(\boldsymbol{r}_p^{\text{worst},t}, \tilde{\mathcal{S}}^t(\boldsymbol{p}^t)) + BLD^{\frac{\alpha}{2}} h_T^{-\alpha} > u(\underline{\boldsymbol{\mu}}_p^t, \tilde{\mathcal{S}}^t(\boldsymbol{p}^t)) - H(t) \right\}$$
$$= \left\{ u(\boldsymbol{r}_p^{\text{worst},t}, \tilde{\mathcal{S}}^t(\boldsymbol{p}^t)) > u(\underline{\boldsymbol{\mu}}_p^t, \tilde{\mathcal{S}}^t(\boldsymbol{p}^t)) - BLD^{\frac{\alpha}{2}} h_T^{-\alpha} - H(t) \right\} \tag{32}$$

Therefore, using (30), (31) and (32), the probability of event $E_3$ is bounded by

$$\text{Prob}\{E_3\}$$
$$\le \text{Prob}\left\{ W^t, u(\boldsymbol{r}_p^{\text{best},t}, G) \ge u(\boldsymbol{r}_p^{\text{worst},t}, \tilde{\mathcal{S}}^t(\boldsymbol{p}^t)), \right.$$
$$u(\boldsymbol{r}_p^{\text{best},t}, G) < u(\bar{\boldsymbol{\mu}}_p^t, G) + BLD^{\frac{\alpha}{2}} h_T^{-\alpha} + H(t)$$
$$\left. u(\boldsymbol{r}_p^{\text{worst},t}, \tilde{\mathcal{S}}^t(\boldsymbol{p}^t)) > u(\underline{\boldsymbol{\mu}}_p^t, \tilde{\mathcal{S}}^t(\boldsymbol{p}^t)) - BLD^{\frac{\alpha}{2}} h_T^{-\alpha} - H(t) \right\}. \tag{33}$$

We want to find a condition under which the probability for $E_3$ is zero. For this purpose, it is sufficient to show that the probability for the right-hand side in (33) is zero. Suppose that the following condition is satisfied:

$$2H(t) + 2BLD^{\frac{\alpha}{2}}h_T^{-\alpha} \leq At^{\theta} \tag{34}$$

Since $G \in \mathcal{L}^t(\boldsymbol{p}^t)$, we have $u(\underline{\boldsymbol{\mu}}_p^t, \tilde{\mathcal{S}}^t(\boldsymbol{p}^t)) - u(\bar{\boldsymbol{\mu}}_p^t, G) \geq At^{\theta}$, which together with (34) implies that

$$u(\underline{\boldsymbol{\mu}}_p^t, \tilde{\mathcal{S}}^t(\boldsymbol{p}^t)) - u(\bar{\boldsymbol{\mu}}_p^t, G) - \left(2H(t) + 2BLD^{\frac{\alpha}{2}}h_T^{-\alpha}\right) \geq 0 \tag{35}$$

Rewriting yields

$$u(\underline{\boldsymbol{\mu}}_p^t, \tilde{\mathcal{S}}^t(\boldsymbol{p}^t)) - H(t) - BLD^{\frac{\alpha}{2}}h_T^{-\alpha} \geq u(\bar{\boldsymbol{\mu}}_p^t, G) + H(t) + BLD^{\frac{\alpha}{2}}h_T^{-\alpha} \tag{36}$$

If (36) holds true, the three components of the right-hand side in (33) cannot be satisfied at the same time: Combining the second and third component of (33) with (36) yields $u(\boldsymbol{r}_p^{\text{best},t}, G) < u(\boldsymbol{r}_p^{\text{worst},t}, \tilde{\mathcal{S}}^t(\boldsymbol{p}^t))$, which contradicts the first term of (33). Therefore, under condition (34), it follows that $\text{Prob}\{E_3\} = 0$.

So far, the analysis was performed with respected to an arbitrary $H(t) > 0$. In the remainder of the proof, we choose $H(t) = Br^{\max}t^{-z/2}$. Then, using (22) and (23), we have

$$\text{Prob}\{E_1\} \leq B\exp\left(\frac{-2H(t)^2t^z\log(t)}{B^2(r^{\max})^2}\right) \leq B\exp\left(-2\log(t)\right) \leq Bt^{-2} \tag{37}$$

and analogously

$$\text{Prob}\{E_2\} \leq Bt^{-2} \tag{38}$$

To sum up, under condition (34), using (20), the probability in (19) is bounded by

$$\text{Prob}\left\{V_G^t, W^t\right\} \leq \text{Prob}\left\{E_1 \cup E_2 \cup E_3\right\} \leq \text{Prob}\left\{E_1\right\} + \text{Prob}\left\{E_2\right\} + \text{Prob}\left\{E_3\right\} \leq 2Bt^{-2}$$

Given this we have:

$$\begin{aligned}
\mathbb{E}[R_s(T)] &\leq (1 - \frac{1}{e})Br^{\max} \times \sum_{t=1}^{T}\sum_{G \in \mathcal{L}^t(\boldsymbol{p}^t)} \text{Prob}\left\{V_G^t, W^t\right\} \\
&\leq (1 - \frac{1}{e})Br^{\max}\binom{M^{\max}}{B}\sum_{t=1}^{T}2Br^{\max}t^{-2} \\
&\leq (1 - \frac{1}{e})B^2(r^{\max})^2\binom{M^{\max}}{B} \cdot 2\sum_{t=1}^{\infty}t^{-2} \\
&\leq (1 - \frac{1}{e})B^2(r^{\max})^2\binom{M^{\max}}{B}\frac{\pi^2}{3} \tag{39}
\end{aligned}$$

where $\binom{M^{\max}}{B}$ is maximum possible number of subsets of size $B$ in each time slot. $\qquad\square$

Now we give a bound for $\mathbb{E}[R_n(T)]$.

**Lemma 4** (Bound for $\mathbb{E}(R_n(T))$). *Let $K(t) = t^z\log(t)$ and $h_T = \lceil T^{\gamma}\rceil$, where $0 < z_n < 1$ and $0 < \gamma < \frac{1}{D}$. If the algorithm is run with these parameters, Assumption 1 holds true, the regret $\mathbb{E}[R_n(T)]$ is bounded by*

$$\mathbb{E}[R_n(T)] \leq 3Br^{\max}LD^{\frac{\alpha}{2}}T^{1-\gamma\alpha} + \frac{A}{1+\theta}T^{1+\theta} \tag{40}$$

*Proof of Lemma 4.* For $1 \leq t \leq T$, consider the event $W^t$ as in the previous proof, the regret due to near-optimal subsets can be written as

$$R_n(T) = \sum_{t=1}^{T}I_{\{W^t, \mathcal{S}^t \in \mathcal{S}_b \setminus \mathcal{L}^t(\boldsymbol{p}^t)\}} \times \left((1 - \frac{1}{e}) \cdot u\left(\boldsymbol{r}^t, \mathcal{S}^{*,t}(\boldsymbol{x}^t)\right) - u\left(\boldsymbol{r}^t, \mathcal{S}^t\right)\right) \tag{41}$$

where in each time slot in which the selected subset $\mathcal{S}^t$ is near-optimal, i.e., $\mathcal{S}^t \in \mathcal{S}_b \backslash \mathcal{L}^t(\boldsymbol{p}^t)$, the regret is considered for selecting $\mathcal{S}^t$ instead of the $\mathcal{S}^{*t}(\boldsymbol{x}^t)$. Let $Q^t = W^t \cap \{\mathcal{S}^t \in \mathcal{S}_b \backslash \mathcal{L}^t(\boldsymbol{p}^t)\}$ denotes the event of selecting a near-optimal arm set. Then, we have

$$\mathbb{E}\left[R_n(T)\right] = \sum_{t=1}^{T} \mathbb{E}\left[I_{\{Q^t\}} \times \left((1-\frac{1}{e}) \cdot u\left(\boldsymbol{r}^t, \mathcal{S}^{*,t}(\boldsymbol{x}^t)\right) - u\left(\boldsymbol{r}^t, \mathcal{S}^t\right)\right)\right]$$

By the definition of conditional expectation, this is equivalent to

$$\mathbb{E}\left[R_n(T)\right] = \sum_{t=1}^{T} \mathrm{Prob}\{Q(t)\} \cdot \mathbb{E}\left[(1-\frac{1}{e}) \cdot u\left(\boldsymbol{r}^t, \mathcal{S}^{*,t}(\boldsymbol{x}^t)\right) - u\left(\boldsymbol{r}^t, \mathcal{S}^t\right) \mid Q(t)\right]$$

$$\leq \sum_{t=1}^{T} \mathbb{E}\left[(1-\frac{1}{e}) \cdot u\left(\boldsymbol{r}^t, \mathcal{S}^{*,t}(\boldsymbol{x}^t)\right) - u\left(\boldsymbol{r}^t, \mathcal{S}^t\right) \mid Q(t)\right]$$

Now, let $t$ be the time slot, where $Q(t)$ holds true, i.e., the algorithm enters an exploitation phase and $J \in \mathcal{S}_b \backslash \mathcal{L}^t(\boldsymbol{p}^t)$. By the definition of $\mathcal{P}^{\mathrm{ue},t}$, it holds that $C^t(p_m^t) > K(t) = t^z \log(t)$ for all $p_m^t \in \boldsymbol{p}^t$. In addition, since $J \in \mathcal{S}_b \backslash \mathcal{L}^t(\boldsymbol{p}^t)$, it holds

$$u(\boldsymbol{\mu}_p^t, \tilde{\mathcal{S}}^t(\boldsymbol{p}^t)) - u(\bar{\boldsymbol{\mu}}_p^t, J) < At^\theta \tag{42}$$

To bound the regret, we have to give an upper bound on

$$\sum_{t=1}^{T} \mathbb{E}\left[(1-\frac{1}{e}) \cdot u\left(\boldsymbol{r}^t, \mathcal{S}^{*,t}(\boldsymbol{x}^t)\right) - u\left(\boldsymbol{r}^t, J\right) \mid Q(t)\right]$$

$$= \sum_{t=1}^{T} \left((1-\frac{1}{e}) \cdot u\left(\boldsymbol{\mu}^t, \mathcal{S}^{*,t}(\boldsymbol{x}^t)\right) - u\left(\boldsymbol{\mu}^t, J\right)\right) \tag{43}$$

Applying Hölder condition several times yields:

$$(1 - \frac{1}{e}) \cdot u\left(\boldsymbol{\mu}^t, \mathcal{S}^{*,t}(\boldsymbol{x}^t)\right) - u\left(\boldsymbol{\mu}_x^t, J\right)$$

$$\leq (1 - \frac{1}{e}) \cdot u\left(\tilde{\boldsymbol{\mu}}_p^t, \mathcal{S}^{*,t}(\boldsymbol{x}^t)\right) + BLD^{\frac{\alpha}{2}} h_T^{-\alpha} - u\left(\boldsymbol{\mu}_x^t, J\right)$$

$$\leq (1 - \frac{1}{e}) \cdot u\left(\tilde{\boldsymbol{\mu}}_p^t, \mathcal{S}^{*,t}(\boldsymbol{p}^t)\right) + BLD^{\frac{\alpha}{2}} h_T^{-\alpha} - u\left(\boldsymbol{\mu}_x^t, J\right)$$

$$\leq u\left(\tilde{\boldsymbol{\mu}}_p^t, \mathcal{S}^t(\boldsymbol{p}^t)\right) + BLD^{\frac{\alpha}{2}} h_T^{-\alpha} - u\left(\boldsymbol{\mu}_x^t, J\right)$$

$$\leq u\left(\boldsymbol{\mu}_p^t, \mathcal{S}^t(\boldsymbol{p}^t)\right) + 2BLD^{\frac{\alpha}{2}} h_T^{-\alpha} - u\left(\boldsymbol{\mu}_x^t, J\right)$$

$$\leq u\left(\boldsymbol{\mu}_p^t, \mathcal{S}^t(\boldsymbol{p}^t)\right) + 3BLD^{\frac{\alpha}{2}} h_T^{-\alpha} - u\left(\bar{\boldsymbol{\mu}}_p^t, J\right)$$

$$\leq 3BLD^{\frac{\alpha}{2}} h_T^{-\alpha} + At^\theta$$

where the third inequality follows the definition of $\tilde{\mathcal{S}}^{*,t}(\boldsymbol{p}^t)$ and $\tilde{\mathcal{S}}^t(\boldsymbol{p}^t)$. Using $h_T^{-\alpha} = \lceil T^\gamma \rceil^{-\alpha} \leq T^{-\gamma\alpha}$, we further have

$$\mathbb{E}\left[u\left(\boldsymbol{r}^t, \mathcal{S}^{*,t}(\boldsymbol{x}^t)\right) - u\left(\boldsymbol{r}^t, J\right) \mid Q(t)\right] \leq 3BLD^{\frac{\alpha}{2}} T^{-\alpha\gamma} + At^\theta \tag{44}$$

Therefore, the regret can be bounded by

$$\mathbb{E}[R_n(T)] \leq \sum_{t=1}^{T} \left(3BLD^{\frac{\alpha}{2}} T^{-\alpha\gamma} + At^\theta\right) \leq 3BLD^{\frac{\alpha}{2}} T^{1-\alpha\gamma} + \frac{A}{1+\theta} T^{1+\theta}. \tag{45}$$

$\square$

The over all regret is now bounded by applying the above Lemmas.

*Proof of Theorem 1.* First, let $K(t) = t^z \log(t)$ and $h_T = \lceil T^\gamma \rceil$, where $0 < z < 1$ and $0 < \gamma < \frac{1}{D}$; let $H(t) = Br^{\max} t^{-z/2}$; let the condition $2H(t) + 2BLD^{\frac{\alpha}{2}} T^{-\alpha\gamma} \leq At^\theta$ be satisfied for all

$1 < t < T$. Combining the results of Lemma 1, 2, 3, the regret $R(T)$ is bounded by

$$R(T) \leq (1 - \frac{1}{e}) \cdot Br^{\max}2^D \left( \log(T)T^{z+\gamma D} + T^{\gamma D} \right) + (1 - \frac{1}{e}) \cdot B^2 r^{\max} \binom{M^{\max}}{B} \frac{\pi^2}{3}$$

$$+ 3BLD^{\frac{\alpha}{2}}T^{1-\alpha\gamma} + \frac{A}{1+\theta}T^{1+\theta}$$

The summands contribute to the regret with leading orders $O(\log(T)T^{z+\gamma D})$, $O(T^{1-\gamma\alpha})$ and $O(T^{1+\theta})$. In order to balance the leading orders, we select the parameters $z, \gamma, A, \theta$ as following values $z = \frac{2\alpha}{3\alpha+D} \in (0,1), \gamma = \frac{z}{2\alpha} \in (0, \frac{1}{D}), \theta = -\frac{z}{2}$, and $A = 2Br^{\max} + 2BLD^{\alpha/2}$. Note that the condition (34) is satisfied with these values. The the regret $R(T)$ reduces to

$$R(T) \leq (1 - \frac{1}{e}) \cdot Br^{\max}2^D \left( \log(T)T^{\frac{2\alpha+D}{3\alpha+D}} + T^{\frac{D}{3\alpha+D}} \right)$$

$$+ (1 - \frac{1}{e}) \cdot B^2 r^{\max} \binom{M^{\max}}{B} \frac{\pi^2}{3} + \left( 3BLD^{\alpha/2} + \frac{2Br^{\max} + 2BLD^{\alpha/2}}{(2\alpha+D)/(3\alpha+D)} \right) T^{\frac{2\alpha+D}{3\alpha+D}}$$

Then the leading order is $O\left( (1 - \frac{1}{e})Br^{\max}2^D T^{\frac{2\alpha+D}{3\alpha+D}} \log(T) \right)$. □

# B Simulation Extension

## B.1 Regret Analysis

In Fig. 7, we explicitly depict the regret achieved by CC-MAB and other benchmarks. It can be observed clearly that CC-MAB is able to achieve a sublinear regret as proved in Theorem 1. By contrast, the regrets of other benchmarks grow dramatically over time.

Figure 7: Regret analysis.

## B.2 Impact of Arm Arrival

The arm arrival pattern in the previous simulations is determined by real dataset as shown in Fig. 8. To verify the impact of stochastic arm arrival on CC-MAB, we add a data pre-processing procedure to change the arm arrival pattern. Specifically, we assume that the number of arms arriving at each slot follows a normal distribution $\mathcal{N}(\mu_{\mathrm{arm}}, \sigma_{\mathrm{arm}}^2)$, where $\mu_{\mathrm{arm}}$ is set to 50 according to the analysis on Yelp data. The parameter $\sigma_{\mathrm{arm}}$ is used to control the stochasticity of the arm arrival pattern. Fig. 9 shows the performances achieved by Oracle and CC-MAB under various arm arrival patterns. It can be observed that CC-MAB works well under all three considered arm arrival patterns.

Figure 8: Real arm arrival pattern.

Figure 9: Impact on stochastic arm arrival.