[Reviews · NeurIPS 2018]

Reviewer 1



The paper introduces a novel algorithm with regret guarantees for stochastic contextual combinatorial multi-armed bandits for volatile arms and submodular rewards. The dependence on dimension of the space of their regret bound is not great and in general constants seem large. A discussion of these quantities is warranted here. The main drawback is that there is no comparison to other bounds. It seems that they can instantiate their algorithm to simpler settings that have been analyzed by previous work. How does their bound compare to the regret of previous algorithms in these simpler setting? Do they have any guess at the lower bound? Splitting the context space by hypercubes seems expensive. Is the algorithm computationally efficient? Please add a discussion on this. As for the experiments, it would have been nice to have seen results for more than one dataset. I understand their reasoning for their definition of regret, but is it necessary in their analysis to include the factor (1-1/e)?

Reviewer 2



This paper studies the stochastic contextual combinatorial multi-armed bandit with volatile arms (i.e. arms can become available or unavailable in each round) and submodular rewards (i.e the reward observed for the selected arms is not a sum of their individual rewards, but a diminishing function of their returns, dependent on the relationship between selected arms). The authors very clearly introduce the reader to the problem and to the key relevant topics (the contextual, combinatorial and volatile bandits, and submodular functions). They successfully explain their main contribution (a contextual combinatorial multi-armed bandit algorithm compatible with submodular reward functions and volatile arms), by explicitly stating the novelty of their work: "[17] considers a special submodular reward function that depends on a linear combination of individual rewards and assumes a constant set of arms which is very different from the volatile arms in our work". As such, the significance of their work is important, since they consider (a) general submodular reward functions (as in equation (1)), and (2) allow for arms to dynamically become available or unavailable, all within the contextual-combinatorial bandit setting. Their proposed algorithm is based on ideas from the contextual bandit with similarity information literature. They divide the arms into different groups based on their context, learn the expected quality/reward for each group of arms, and assume that arms with similar context have similar properties (as in Lipschitz bandits via the Holder condition). The authors propose a heuristic to explore under-explored groups (hypercubes in space), in combination with the greedy algorithm (1) to solve the submodular function maximization. The key parameters of their algorithm are the hypercube partition sizes $h_T$ and the control policy $K(t)$, which they specify in connection with their theoretical regret bound. Some insights into the tightness of the bound, and its connection to the stochastic pattern of the arm arrival are provided. The authors have provided in their rebuttal further comments on the impact of context space dimension in the regret bound, which would be interesting to add to the final manuscript. However, it is unclear how realistic the budget assumption in corollary 1 is. The authors argue in their rebuttal that it is realistic in some practical applications (such as crowdsourcing). However, it is likely that in other applications, having a smaller budget than the available arms at every time instant would not apply. Thus, it would be interesting if the authors include such discussion on their final manuscript. Nevertheless, further analysis (both theoretical and empirical) of realistic arm arrival patterns would increase the impact of their work. The evaluation section focuses on a real problem of crowdsourcing, which is specifically re-casted to fit their problem formulation. They conclude that (a) contextual-combinatorial bandits perform better than non-contextual ones, (2) not considering submodularity incurs an obvious reward loss (because the underlying reward is indeed submodular); and (3) increasing the budget reduces regret (no learning required). These results are not surprising and in some sense expected, since the problem set up is specifically targeted to the designed framework. On the one hand, the authors have agreed to extend the evaluation of their algorithm to longer runs (as only 200 time slots are shown in Figure 3). By doing so, they can empirically evaluate the presented regret bound (as it is not clear from the provided figures whether it is sublinear, or how it compares to the theoretical bound). On the other, I would still encourage the authors to consider simulated scenarios where they can empirically evaluate the impact of stochastic pattern of the arm arrivals. Finally, their algorithm is based on static partitions of the space (hypercubes $h_t$). I thank the authors for the insights provided in their rebuttal regarding (a) the incurred algorithm complexity and (b) extending the partition to be non-homogeneous and/or dynamic. As such, I encourage the authors to add them to the final manuscript.

Reviewer 3



This paper studies the contextual combinatorial multi-armed bandit (MAB) problem, where the goal is to select in each round the subset of arms that maximize the expected cumulative reward. The main contribution of the paper is the formulation and analysis of the CC-MAB algorithm, which addresses this contextual combinatorial MAB problem for the setting where the reward function is submodular and the set of arms can change between rounds. Although the algorithm builds on previously well-studied MAB frameworks, it has a relatively significant contribution by bringing together the combinatorial MAB setting, with the use of contexts, and with the challenging setting of "volatile arms". The paper is well written and the studied problem is interesting and quite well motivated for real-world applications. On the other hand, the experiments seem a bit lacking. I wonder why there was no empirical comparison to combinatorial MAB algorithms (for instance, from references [5,7])? Also, none of the considered benchmarks considers the submodularity of the reward function, thus, as it is pointed out in the paper, they incur obvious reward loss. More details about the comparison of CC-MAB behavior compared to submodular MAB algorithms (for instance, the one proposed in [Chen, Krause, Karbasi, NIPS 2017]) would be appreciated. Minor comments/typos: - line 172: performs *the* following steps? - line 221: budge -> budget ======================= Post author response edit: I have read the rebuttal and thank the authors for addressing some of my concerns.